# Dislocation Density Based Flow Stress Model Applied to the PFEM Simulation of Orthogonal Cutting Processes of Ti-6Al-4V

**DOI:** 10.3390/ma13081979

**Published:** 2020-04-24

**Authors:** Juan Manuel Rodríguez, Simon Larsson, Josep Maria Carbonell, Pär Jonsén

**Affiliations:** 1Department of Mechanical Engineering, EAFIT University, Medellin 050022, Colombia; 2Division of Mechanics of Solid Materials, Luleå University of Technology, 97187 Luleå, Sweden; simon.larsson@ltu.se (S.L.); par.jonsen@ltu.se (P.J.); 3Department of Engineering, Faculty of Science and Technology, Universitat de Vic, 08500 Vic, Spain; cpuigbo@cimne.upc.edu; 4Centre Internacional de Mètodes Numèrics en Engninyeria (CIMNE), 08034 Barcelona, Spain

**Keywords:** Particle Finite Element Method (PFEM), metal cutting processes, Johnson-Cook, dislocation density constitutive model, titanium Ti6Al4V

## Abstract

Machining of metals is an essential operation in the manufacturing industry. Chip formation in metal cutting is associated with large plastic strains, large deformations, high strain rates and high temperatures, mainly located in the primary and in the secondary shear zones. During the last decades, there has been significant progress in numerical methods and constitutive modeling for machining operations. In this work, the Particle Finite Element Method (PFEM) together with a dislocation density (DD) constitutive model are introduced to simulate the machining of Ti-6Al-4V. The work includes a study of two constitutive models for the titanium material, the physically based plasticity DD model and the phenomenology based Johnson–Cook model. Both constitutive models were implemented into an in-house PFEM software and setup to simulate deformation behaviour of titanium Ti6Al4V during an orthogonal cutting process. Validation show that numerical and experimental results are in agreement for different cutting speeds and feeds. The dislocation density model, although it needs more thorough calibration, shows an excellent match with the results. This paper shows that the combination of PFEM together with a dislocation density constitutive model is an excellent candidate for future numerical simulations of mechanical cutting.

## 1. Introduction

Metal machining is an important manufacturing technique in major mechanical industries like the automobile or aerospace. Using this technology, the material is removed continuously from the work piece by a cutting tool, creating a new shape for the piece. Although new manufacturing techniques, such as additive manufacturing [1], have recently been developed for metal processing, machining is the most prevalent manufacturing operation in terms of volume and expenditure. It has been estimated that machining expenditure contributes to approximately 5% of the GDP in developed countries [2,3]. However, still today, after decades of research, machining presents a challenging and exciting intellectual problem for researchers and practitioners.

Numerical simulation of metal cutting processes has become important as it is capable to provide detailed insight of the processes. Via the improvement of the accuracy of the numerical simulations it will be the possibility to replace difficult, costly or even impossible experiments used for tool and process design. Also, the modelling and simulation at the design stage can contribute to the shortening of the development time. It will reduce the costs for the design of new cutting tools and the production of new components.

To properly model cutting processes a numerical tool should include capabilities such as: the treatment of the thermo-mechanical coupling, the reproduction of the material response, the treatment of the incompressibility, and the modelling of the thermo-mechanical contact and friction. It must also include adaptive discretizations to deal with large strain and deformations, the breakage and separation of the chip and prediction of the tool wear.

The finite element method (FEM) is the standard tool to simulate machining processes either in its Lagrangian, Eulerian or Arbitrary Lagrangian Eulerian (ALE) formulation [4,5,6,7,8,9,10,11,12]. As large deformations are present in the metal cutting process, when a Lagrangian mesh is excessively distorted it does not allow the continuation of the simulation. The problem of FE-mesh distortion, is usually solved via mesh adaptivity and automatic remeshing, however, the use of remeshing can introduce problems with diffusion of the state variables.

During the last years, new revolutionary techniques appeared extending and improving the capabilities of FEM in order to model metal cutting. New developed methods can be classified into: methods supported by a mesh, mesh free methods and particle methods. The methods supported by a mesh include the Multi-material Eulerian Method (MMEM) [13,14], the Volume of Solid (VOS) [15], and the Material Point Method (MPM) or Point in Cell (PiC) method [16,17]. Among the particles/meshless methods applied to metal cutting we found: the Smoothed Particle Hydrodynamics (SPH) [18,19], the Finite Point Method (FPM) [20], the Constrained Natural Element Method (CNEM) [21], the Discrete Element Method (DEM) [22], the Maximum Entropy MeshFree method [23], the stabilized optimal transportation meshfree method (OTM) [24], and the Particle Finite Element Method (PFEM) [25,26,27,28]. Most of these new methods are applied using similar simplified models. For instance, they usually consider the tool as rigid with no heat transfer between the tool and the piece [6]. This represents a poor approach if one wants to model realistic chips. Furthermore, these methods have a relatively high computational cost in comparison to FEM [6]. However, some of the new methods, like the PFEM, allow modeling the deformations and the heat transfer between the tool and the chip. With the advantage that the computational cost of the PFEM is similar to that of specialized FEM programs [29].

There are several references [25,26,27,28] that present the capabilities of the PFEM as a numerical tool to model large deformations surpassing the problems of mesh distortions and diffusion of state variables that are typical in FEM simulations. Also, PFEM includes capabilities to deal with thermo-mechanical coupling, contact and friction, treatment of the incompressibility constraint due to plasticity. This numerical method is the better candidate for the purposes of this work. Details about PFEM in metal cutting are available in Reference [25].

The constitutive model is one of the most important features in an accurate modeling of metal cutting processes. For metal cutting simulations a constitutive model must represent the behavior of the material at high strain rates and be able to account for strain softening, strain hardening and thermal softening over a wide range of temperature, strain and strain rates. In many numerical simulations carried out up to now, the material behaviour of metals is represented by phenomenological power laws [7], described in terms of strains and strain rate. Power laws equations relate the flow stress to the plastic strain, strain rate and temperature, but they are not based on the underlying physics of the material deformation. A very common example of a power law is the Johnson-Cook (JC) model [30], the most used constitutive model for the numerical modeling of machining processes [7]. However, one weakness of these empirical relationships is that the precision and accuracy in the prediction of experimental results is inadequate outside the calibration range. Therefore, it is preferable to use models which are related to the physics of the deformation because the range of validity out of the experimental calibration range is larger compared to phenomenological models. There is evidence of this better behavior when the dislocation density (DD) constitutive model was used to model metal cutting with SANMAC 316L stainless steel in References [27,31,32]. A physically based Voyiadjis-Abed model including the hardening due to dynamic strain aging is presented in References [33,34], it is expected that including the dynamics strain aging in the numerical modeling of metal cutting process will increase the accuracy of the results.

A FEM model of orthogonal cutting of Al6061-T6 alloys at high speeds is presented in Reference [35]. The previous model predicts dislocation densities and grain size during machining processes. They found that low cutting speeds and negative rake angles promotes dislocation density and refines the grain size, high cutting speed and positive rake angles generates thin deformation layers in the machined surface. Another FEM model that study the effects of grooved tools on dislocations density and grain size for Al6061-T6 alloy is developed in Reference [36]. They found that groove width change grain size and dislocation density profile along the chip and bellow the machined surface. A coupled Eulerian–Lagrangian (CEL) finite element model with a dislocation density-based material plasticity model was developed in to simulate grain refinement and the evolution of dislocation during orthogonal cutting of commercially pure titanium (CP Ti). They show that the developed model captures the material mechanical behavior and accurately predicts a grain size of CP Ti chips at a cutting speed of 10 mm/s. A dislocation density model of OFHC copper for high-speed cutting was investigated through experiments and FEM simulations in Reference [37]. This study found that the dislocation densities around the tool-chip interface decrease when the cutting speed is increased and the oscillations in cutting forces at high cutting speed is induced by the evolution and distributions of dislocations density. In A dislocation density based viscoplastic constitutive model for lead free solder under drop impact [38]. The constitutive behavior (flow stress) and microstructure evolution were treated independently (uncoupled) in most of the previous dislocations models, and the material behaviors were not influenced by the microstructure evolution. In the present work a coupled model between microstructure and mechanical behavior that describe the behavior of Ti6Al4V during machining that was presented in References [39,40] is used.

Other works have been devoted to modeling tool displacement during ball end milling and its influence on the machined surface roughness [41,42]. The coupling of a FEM/PFEM model, a constitutive behaviour, a microstructure evolution model and tool displacement model are expected to improve the predictability of the developed model.

The present study combines the PFEM [26] and the DD plasticity model of Ti6Al4V presented in References [39,40]. The objective is to improve the numerical approximation of the results and solve the typical problems that appear in the FEM simulation of metal cutting processes. The introduction of the DD model is essential for this purpose. To validate the approach, the today’s most used constitutive model for machining simulations, the phenomenological JC plasticity model, will be taken as reference for comparison. The adaptation of the use of a DD constitutive model into the new advances of the PFEM technique and the validation of the obtained results are the main novelties of this work. These set of improvements will configure an new standard for the future simulation of machining processes.

This paper is structured as follows: In Section 2 the PFEM is described including the basic steps, the meshing procedure and transfer of information. Section 2 makes special emphasis on the improvements made to the PFEM that allow to simulate the chip formation during metal cutting. Then, the JC strength model and a DD constitutive model are presented in Section 3. the experimental measurements carried out in this work are presented in Section 4. In Section 5, the numerical simulation procedures are presented and explained. A set of representative numerical simulations using both previously mentioned constitutive models are presented and analyzed in Section 6, which is followed by a discussion of the results in Section 7 and the conclusions in Section 8.

## 2. The Particle Finite Element Method

The PFEM was developed originally for the treatment of fluids and free surfaces flows [43]. The aim was to utilize a Lagrangian description of motion, that follows and tracks the fluid particles along the domain, in order to characterize fluid free surfaces. The PFEM emerges also to explore the possibilities of using a Lagrangian description of the fluid domains in the modelling of fluid structure interaction, traditionally used to study the problems in solid mechanics. Nowadays, the PFEM is applied in the modeling of a wide range of engineering applications, such as: fluid structure interaction in port engineering, erosion processes in rivers, mixing processes, fluid coupling with thermal effects, slurry in stirred media mills, the pulp fluid in tumbling mills and granular flows [44,45,46,47,48,49]. An early approach for the modelling of cutting, riveting and excavation problems is presented in References [25,50,51,52].

In this paper, the PFEM is adapted and improved for the modelling of machining processes. In solid mechanics, the Lagrangian description is commonly used to model the continuum and, in the specific case of our work, to model machining of material. However, when the formulation is applied using the standard finite element method, it has some critical limitations. The main drawback is that excessively distorted elements prevent simulation progress. In order to surpass this drawback, the particle and Lagragian nature of the PFEM is used in this work.

The original idea of the PFEM was to improve the mesh quality with the re-tessellation of the domain only when needed. This limits the geometry of the elements to triangles in 2D and tetrahedra in 3D. It is common to use only linear elements for simplicity of the operations. This re-tessellation is performed locally which allows the representation of large deformations problems in the continuum domain while global remeshing and transfer of information from mesh to mesh is avoided. Usually, the re-triangulation is performed only when some criteria associated to element distortion is fulfilled. In two dimensions, the re-triangulation consists in recalculating the element connectivity using a *Delaunay* triangulation [53,54] with the position of the particles (i.e., of the mesh nodes) in the updated configuration. The *Delaunay* triangulation maximize the minimum angle of all the triangles in the mesh and thus mesh distortion is minimized, as a consequence the number of triangles with poor aspect ratio in the mesh are minimized or in the better cases removed. An example of the remeshing scheme used in the PFEM simulation of metal cutting processes is shown in Figure 1.

### 2.1. Basic Steps of the PFEM

The continuum in the PFEM is represented using a updated Lagrangian formulation. This description uses an incremental update of the body configuration. That is, all variables are assumed to be known at the beginning of the time step at time *t*. The new set of variables is sought for in the next or updated configuration at time t+Δt. The FEM is used to solve the balance equations that comes from continuum mechanics. Therefore, a mesh is generated in the domain of interest to solve differential equations in the standard form of finite elements.

Recall that in a Lagrangian framework the nodes that represent the domain of interest are treated as material particles for which motion is tracked. This strategy is mandatory to represent the separation of particles from the main domain, and to follow their subsequent motion as individual particles with an initial velocity and acceleration and a known density, under the influence of a gravitational field. In fluid mechanics, the particles that separate represent drops and in machining this methodology is used to study the formation of discontinuous chips.

For completeness, the PFEM process is outlined starting with a set of particles *C* representing the domain of interest, the volume *V* defining the continuum and the mesh *M* that connects the particles in the domain (see Figure 1). A typical solution with the PFEM involves the following steps:Definition of the domain(s) $\Omega_{n}$ in the last converged configuration, $t=t_{n}$, keeping the existing spatial discretization $M_{n}$.Elimination of existing connectivities and reconstruction of the mesh through a *Delaunay* triangulation [53,54] of the domain.Definition of the boundary applying geometrical techniques like the α-shape method [55].Application of a contact search to recognize self-contact and contact between multiple bodies.Transfer of the historical internal variables information to the new discretization $M_{n+1}$Solution of the non-linear system of equations for $t_{n+1}=t_{n}+\Delta t$.Return to step 1 and repeat the process for the next time step.

The above procedure has to be improved for the modelling of metal cutting problems. Some of the weaknesses to be solved are:Using α-shapes for the detection of the external boundaries is not accurate.An improper identification of the boundary can artificially increase or decrease chip thickness.A reconnection of the previous cloud of particles produces a bad discretization of the domain.Particles move as the material deforms and it may happen that in some regions the number of particles stack, the other way around, in other regions particles becomes too low. It harms the accuracy of the solution.

To surpass these obstacle we propose the use of a constrained *Delaunay* triangulation [53,54] together with the introduction and elimination of particles inside the domain. The procedure is designed to preserve the external and internal boundaries and to distribute particles accordingly to the demand of mesh accuracy. The addition and removal of particles enhance the quality of the solution, and allows for the resolution of the different scales of the solution.

The original PFEM removes and adds particles by comparing with a problem dependent characteristic distance *h*. The size of *h* is usually constant throughout the domain (see References [56]). In the present work the criteria for the removal and the addition of particles is extended considering different scenarios:

#### 2.1.1. Removal of Particles

If the distance between two particles $d_{\mathrm{nodes}}$ is less than a certain characteristic distance $h_{\mathrm{rem}}$ ($d_{\mathrm{nodes}}<h_{\mathrm{rem}}$), one of the particles will be removed (see Figure 2a). A similar criterion is applied to the domain boundary $d_{\mathrm{nodes}}<h_{s,\mathrm{rem}}$, being $h_{s,\mathrm{rem}}$ characteristic distance for the boundary (see Figure 2c).If the error estimator $∥\sigma -\sigma_{h}∥$ (see Zienkiewicz and Zhu [57,58]), is smaller than a given tolerance ($∥\sigma -\sigma_{h}∥<\varepsilon_{\mathrm{tol},\mathrm{rem}}$), the node that is in the center of the patch is removed (see Figure 2b). Used error estimators are based on the norm of the isochoric-stress or plastic strain values.

#### 2.1.2. Addition of Particles

If the plastic power generated due to plasticity exceeds a prescribed tolerance, a new particle is introduced at the Gauss point of the finite element $\int_{\Omega^{e}}D_{\mathrm{mech}}\;d\;\Omega^{e}>\varepsilon_{\mathrm{tol},\mathrm{ins}}$ (see Figure 3a).If the radius of an element circumcircle $r_{\mathrm{ec}}$ is greater than a certain characteristic distance $h_{\mathrm{ins}}$ ($r_{\mathrm{ec}}>h_{ins}$), a particle is introduced at the center of the circumcircle (see Figure 3b).If the distance between two particles on the tool tip $d_{\mathrm{nodes}}$ is greater than a certain characteristic distance $h_{s,\mathrm{ins}}$, ($d_{\mathrm{nodes}}>h_{s,\mathrm{ins}}$), a new particle is inserted in the boundary (see Figure 3c).

In the boundary of the domain, the insertion of nodes is fundamental for machining problems. The geometrical description of the tool is done by the introduction of particles on the boundary of the work piece mainly close the tool tip (see Figure 3c). The characteristic distance $h_{s,\mathrm{ins}}$ has a direct relationship with the tool radius size, where the generation of the chip is originated. This allows for the contour increase and creates the shape of the chip.

The insertion of particles takes place in the primary shear zone and in the border where the chip is in contact with the tool. The removal of particles takes place in the formed chip far from the primary shear zone. The leading benefit of the proposed strategy is that an adaptation and mesh improvement of the mesh quality is done through the addition of particles and *Delaunay* triangulation. It is not necessary to create a complete new mesh as in FEM.

The information transfer after performing a new tessellation is crucial in order to conserve the fidelity of the FEM solution from one triangulation to the next. The information necessary in later time steps has to be transferred from the old mesh to the new mesh, it includes the nodal variables like temperatures, pressures, displacements and velocities in the new introduced particles, and internal variables in the new element.

Transfer use to be a very diffusive process. In this work, the transfer is made straight from the previous Gauss point to the Gauss point of the new element. When the mesh does not change and the information is transferred between Gauss points, the equilibrium is preserved before and after the triangulation. When the mesh change, the equilibrium is perturbed locally in areas where introduction and deletion of particles occur and the transformation of the information is unavoidable. A thermo-mechanical test was presented in a previous work (see Reference [26]), where PFEM and FEM solutions were compared. Similar results were obtained, showing that the transfer of the internal variables from Gauss point to Gauss point as is done in the present work has a negligible effect in problems similar to the simulations carried out in this work.

More details about the PFEM in metal cutting can be found in Reference [27].

## 3. Constitutive Models

As stated in the objectives of the article, accurate reproduction of the material response is a critical feature to consider. In a typical machining event, strain rates between $10^{3}$ and $10^{6}$ s${}^{-1}$ and high temperature may be attained within the primary shear zone, while the remainder of the work piece deforms at moderate or low strain rates. In other manufacturing process like extrusion, forging and rolling the maximum strain rate reach $10^{3}$ s${}^{-1}$ [59]. The high strain rates and temperature in machining demonstrates the extreme conditions that take place in that process. In this section, two constitutive models that describe the behavior of metal in a wide range of temperatures, strain and strain rates are presented.

### 3.1. Johnson-Cook Strength Model

The *Johnson-Cook* (JC) strength model [30] is a constitutive model categorized as a phenomenological model. Under certain conditions, the flow stress of some materials was observed to behave in a manner that could be described reasonably well by a particular mathematical formula. The JC model describes the flow of the material as the product of three mathematical terms:(1)$$\sigma_{y}=A\left(1+\frac{B}{A}\;\varepsilon^{n}\right)\left(1+C\ln{\dot{\varepsilon }}^{{}^{\prime }}\right)\left(1-{\theta^{{}^{\prime }}}^{m}\right)$$
composed by *A*, the initial yield strength of the material at room temperature $T_{0}$ and at a reference strain rate of ${\dot{\varepsilon }}_{0}$. ε the equivalent plastic strain, ${\dot{\varepsilon }}^{{}^{\prime }}$ the strain rate non-dimensionalized by the reference strain rate, and *B*, *C*, *m* and *n* are fitting constants.

In Equation (Equation 1), $\theta^{{}^{\prime }}$ is given by
(2)$$\theta^{{}^{\prime }}=\frac{T-T_{0}}{T_{melt}-T_{0}},$$
where $T_{melt}$ is the melting temperature and *T* is the work piece temperature. The non-dimensionalized strain rate ${\dot{\varepsilon }}^{{}^{\prime }}$ is given by
(3)$${\dot{\varepsilon }}^{{}^{\prime }}=\frac{\dot{\varepsilon }}{{\dot{\varepsilon }}_{0}},$$
where $\dot{\varepsilon }$ is the effective plastic strain rate and ${\dot{\varepsilon }}_{0}$ is the reference strain rate.

The JC model (Equation (Equation 1)) assumes an independent effect of the strains, strain rates and the temperatures on the yield strength. In Equation (Equation 1), the expression in the first bracket constitutes a nonlinear strain hardening law, the expression in the second bracket models the strain rate hardening and third bracket represents thermal softening, respectively. In Table 1, a set of JC parameters for Ti6Al4V available in the literature together with the set of JC parameters obtained in the present study are presented. Set no. 7 was obtained using an optimization algorithm (curve fitting) on the experimental stress-strain curves available in Reference [40]. Although the sets of properties of the JC model presented in Table 1 correspond to the Ti6Al4V material, it is found that they differ considerably from each other. Mathematically, there is no guarantee that using the same experimental stress-strain curves, the same JC parameters will be obtained by different authors. In addition, the JC parameters also depend on the experiment that was carried out to obtain the parameters, as well as the maximum temperatures, deformations and deformation rates achieved in these experiments (see Reference [60]).

### 3.2. Dislocation Density Constitutive Model

This section presents a summary of the dislocation density constitutive model developed in References [39,40,67,68]. The flow stress consist of three parts, the long range, the short-range parts of the resistance to the motion of dislocations and the drag component [69,70,71]
(4)$$\sigma_{y}=\sigma_{G}+\sigma^{*}+\sigma_{drag},$$
where $\sigma_{G}$ is the athermal stress contribution from the long-range interactions of the dislocations substructure, $\sigma^{*}$ is the friction stress needed to move dislocations through the lattice and pass short-range obstacles and $\sigma_{drag}$ is the viscous drag component of the flow stress.

The long-range term of Equation (Equation 4) is taken from Reference [72] as,
(5)$$\sigma_{G}=m\alpha Gb\sqrt{\rho_{i}},$$
where *m* is the Taylor orientation factor, α is the proportionality factor, *b* is the Burger’s vector, $\rho_{i}$ is the immobile dislocation density and *G* the shear modulus. The shear modulus *G* can be computed from the Young’s modulus *E* and Poisson ratio ν as
(6)$$G=\frac{E}{2(1+\nu )}.$$
Here, it is assumed that the Young’s modulus *E* depends on the temperature.

The short-range stress component according to Reference [73] can be represented as,
(7)$$\sigma^{*}=\tau_{0}G{\left[1-{\left[\frac{kT}{\Delta f_{0}Gb^{3}}\ln\left(\frac{{\dot{\varepsilon }}^{ref}}{{\dot{\varepsilon }}^{p}}\right)\right]}^{1/q}\right]}^{1/p},$$
where *k* is the Boltzmann constant, *T* is the temperature, $\Delta f_{0}Gb^{3}$ is the activation energy neccesary to overcome short-range obstacles, $\tau_{0}G$ is the shear strength in the absence of thermal energy, ${\dot{\varepsilon }}^{ref}$ is the reference strain rate and $\dot{\varepsilon }$ is the strain rate. The exponents 0 < *p* ≤ 1 and 0 < *q* ≤ 2 are related to the shape of energy barriers.

The part of the flow stress due to drag $\sigma_{drag}$ can be computed based on a formulation proposed in Reference [74] as follows
(8)$$\sigma_{drag}=G\left(\frac{\frac{B_{e}}{B_{p}}+\frac{T}{300}}{C_{drag}}\right){\dot{\varepsilon }}^{p},$$
where $\frac{B_{e}}{B_{p}}$ and $C_{drag}$ are calibration parameters. At strain rates greater than $10^{3}$ the term $\sigma_{drag}$ is described by Equation (Equation 8), at smaller strain rates $\sigma_{drag}=0$.

#### 3.2.1. Evolution of Immobile Dislocation Density

The immobile dislocation density is expressed in terms of hardening (+) and recovery (−) contributions as,
(9)$${\dot{\rho }}_{i}={\dot{\rho }}_{i}^{\left(+\right)}-{\dot{\rho }}_{i}^{\left(-\right)}.$$

The recovery of immobile dislocation density can be obtained as the sum of a term due to glide, a term due to climb and a term due to globularization as
(10)$${\dot{\rho }}_{i}^{\left(-\right)}={\dot{\rho }}_{i(\mathrm{glide})}^{\left(-\right)}+{\dot{\rho }}_{i(\mathrm{climb})}^{\left(-\right)}+{\dot{\rho }}_{i(\mathrm{glob})}^{\left(-\right)}.$$

The increase in immobile dislocations density is assumed to be proportional to the plastic strain rate, which is related to the density of mobile dislocations and inversely to the mean free path
(11)$${\dot{\rho }}_{i}^{\left(+\right)}=\frac{m}{b}\frac{1}{\Lambda }\dot{\varepsilon }$$
the mean free path Λ can be computed from the grain size (*g*) and the subgrain diameter (*s*) as,
(12)$$\frac{1}{\Lambda }=\left(\frac{1}{g}+\frac{1}{s}\right).$$

The evolution of the subgrain diameter has been modeled using a relation proposed by Holt [75].
(13)$$s=\frac{K_{c}}{\sqrt{\rho_{i}}},$$
where $K_{c}$ is a calibration parameter.

The grain growth model can be written according to Reference [76] as,
(14)$$g^{n}-g_{0}^{n}=Kt.$$

Here, $g_{0}$ is the grain size at the beginning of the simulation, *t* is the time elapsed and *K* is a calibration parameter.

The submodel for recovery by glide can be written based on the formulation presented in Reference [77]
(15)$${\dot{\rho }}_{glide}^{\left(-\right)}=\Omega {\dot{\rho }}_{i}{\dot{\varepsilon }}^{p},$$
where Ω is a parameter that depends on the temperature, it is usually called recovery function.

Static recovery controlled by diffusion climb is assumed to have the form according to Reference [67]
(16)$$\rho_{i(\mathrm{climb})}^{\left(-\right)}=2c_{\gamma }D_{app}\frac{c_{v}}{c_{v}^{eq}}\frac{Gb^{3}}{kT}\left(\rho_{i}^{2}-\rho_{eq}^{2}\right).$$

$c_{\gamma }$ is a material coefficient, $D_{app}$ is the apparent diffusivity, $c_{v}$ is the fraction of vacancies, $c_{v}^{eq}$ is the thermal equilibrium vacancy concentration and $\rho_{eq}$ is the equilibrium immobile dislocations density. Mathematical and modeling details about $D_{app}$ term can be found in References [39,40]. The reduction of dislocations density due to globularization is given by
(17)$${\dot{\rho }}_{i(\mathrm{glob})}^{\left(-\right)}=\psi {\dot{X}}_{g}\left(\rho_{i}-\rho_{eq}\right)H\left(\rho_{i}-\rho_{i}^{crit}\right).$$
Here, $\rho_{eq}$ is the equilibrium value of dislocation density, $\rho_{i}^{crit}$ is the critical dislocation density above which globularization is initiated, ${\dot{X}}_{g}$ is the globularization rate, ψ is a calibration constant and *H* is the step function. The reduction of dislocations density due to globularization is active until the dislocations density $\rho_{i}$ is less or equal to the equilibrium dislocation density $\rho_{eq}$, otherwise, ${\dot{\rho }}_{i(\mathrm{glob})}^{\left(-\right)}=0$.

According to [39,40], globularization is modeled as,
(18)$$X_{g}=X_{d}+\left(1-X_{d}\right)X_{s}.$$
Here, $X_{g},X_{d}$ and $X_{s}$ denote total globurized, its dynamic and its static component, respectively.

The rate of static globularization can be written as
(19)$${\dot{X}}_{s}=M\frac{\dot{g}}{g},$$
where $\dot{g}$ is the grain grow rate and *M* is a material parameter. The rate of dynamic globularization is modeled based on Reference [78] as,
(20)$${\dot{X}}_{d}=\frac{Bk_{p}{\left(\varepsilon^{p}\right)}^{k_{p}-1}}{e^{B{\left(\varepsilon^{p}\right)}^{k_{p}}}}{\dot{\varepsilon }}^{p},$$
where $\varepsilon^{p}$ is the equivalent plastic strain, and B and $k_{p}$ are material parameters. The static and dynamic globularization only grow when $\rho_{i}\geq \rho_{i}^{crit}$.

#### 3.2.2. Evolution of Excess of Vacancy Concentration

In Reference [79] a model for excess vacancy concentration with generation and aniquilation components was proposed. That model can be written as
(21)$${\dot{c}}_{v}=\left[\chi \frac{m\alpha Gb^{2}\sqrt{\rho_{i}}}{Q_{vf}}+\zeta \frac{c_{j}}{4b^{2}}\right]\frac{\Omega_{0}}{b}{\dot{\varepsilon }}^{p}-D_{vm}\left[\frac{1}{s^{2}}+\frac{1}{g^{2}}\right]\left(c_{v}-c_{v}^{eq}\right)+c_{v}^{eq}\left(\frac{Q_{vf}}{kT^{2}}\right)\dot{T}.$$
Here, χ is the fraction of mechanical energy spent on vacancy generation, $Q_{vf}$ is the activation energy of vacancy formation, ζ is the neutralization effect by vacancy emiting and absorbing jogs, $c_{j}$ is the concentration of jogs, $\Omega_{0}$ is the atomic volume, $D_{vm}$ is the diffusivity of vacancies and $\dot{T}$ is the time derivative of the temperature field.

The coupled solution of the system of ordinary differential equations (ODE) of Equations (Equation 9) and (Equation 21) is carried out at elementary level with a Newton-Raphson iterative scheme, a similar solution scheme to coupled equations is presented in References [27,31,32]. The coupled solution of Equations (Equation 9) and (Equation 21) gives the new value of $\rho_{i}$ and $c_{v}$. The starting values at the beginning of the cutting simulation of $\rho_{i}$ = $\rho_{i}^{init}$ at 20 °C and of $c_{v}$ = 0. The obtained value of $\rho_{i}$ is used in Equation (Equation 5) to calculate the long-range component of the flow stress. The long-range component is used in the non-linear solution of the mechanical and thermal problem of the next time step. A flowchart about how to implement the dislocations density constitutive model is presented in Box 1 . In Reference [27] extra details about how to implement a dislocations density constitutive model inside a in-house PFEM software are presented. The material parameters of the dislocation density constitutive model are presented in Appendix A.

Box 1Flowchart of the implementation of the dislocations density constitutive model.
Given the new value of ${\dot{\varepsilon }}^{p}$ and $\varepsilon^{p}$, and the old value $X_{d}$, $X_{s}$, $X_{g}$, $\rho_{i}$, *g* and $c_{v}$.
Update $\rho_{i}$ and $c_{v}$ with the non-linear solution of Equations (Equation 9) and (Equation 21). The non-linear solution for the set of equations is carried out a Newton-Raphson scheme.Calculate the updated value of the grain size *g* with Equation (Equation 13).Calculate the grain grow rate with the old, the new value of the grain size and the size of the time step.Update $X_{s}$ with Equation (Equation 19), $X_{d}$ with Equation (Equation 20) and $X_{g}$ with Equation (Equation 18).Calculate the long-range component of the flow stress using Equation (Equation 5).



## 4. Experimental Measurements

In this experimental investigation, triangular inserts TNMG 160408-QF of grade 4015 coated with TiCN-A_2_O_3_-TiN by chemical vapour deposition (CVD) are used. This combination of coating and substrate is preferable for both continuous and intermittent turning operations. The insert was mounted on a turning tool holder PTGNL 3225P having a wedge angle of 90° in order to achieve a clearance angle of 6°. The insert used had a grooved geometry made up by a radius of 46 μm, a primary land of 0.15 mm and a rake face with an active rake face angle of −6° due to the tool holder. The edge geometry have great influence on the cutting and feed force components and measured profile of the cutting edge, which was used to define the simulation model, are shown in Figure 4. A George Fischer CNC turning lathe was used for the turning tests where prefabricated tubes are orthogonally cut in a machining operation. The tubes have one closed end and an outer diameter of 140 mm, a thickness of 3 mm and a length of 115 mm. For measuring the cutting and feed force components, a three component Kistler dynamometer of type 9263 was used together with a 300 Hz low pass filter. The measured force time curves, converges towards a steady state value, similar to the one show Figure 5, Figure 9 and Figure 10. The values presented in Table 5 were obtained by averaging the signal between tool displacement 0.4 mm and 0.8 respectively.

## 5. Simulation Procedures

An implicit PFEM was used with an updated Lagrangian formulation accounting for large deformation, large strains, thermal coupling strains, heat transfer due to conduction, heat generation due to plasticity and heat generation due to friction [26,27]. A staggered/fractional step approach called *isothermal split method* [80,81,82] was used to couple the mechanical and thermal problem. The full Newton–Raphson (NR) method including line search was used to solve the mechanical problem (equilibrium equations). Also, NR was used to solve the nonlinear equations that comes from the thermal problem and from the dislocation density constitutive model [27]. Continuous use of *Delaunay triangulation* [53] and the insertion and removal of particles are the principal ingredients of the PFEM to maintain a reasonable shape of elements and to capture gradients of strain, stress, strain rate and temperature [25].

An orthogonal cutting operation was employed to mimic 2D plane strain conditions. The thickness of the work piece, used for all test cases is equal to 3 mm. Coupled thermo-mechanical plane strain elements are used for the discretization of the work piece and the tool, respectively. A stabilized mixed displacement–pressure-temperature finite element formulation with linear shape functions is used for the work piece [26]. A mixed stabilized formulation employed to avoid problems due to material incompressibility. In the test cases presented in this work, the initial number of particles to describe the work piece is 105, see Figure 6 and Figure 7. The number of particles increases progressively as the simulation advances. The tool geometry is a section of an scanned real tool profile (see Figure 4b) and is discretized by 2298 three-node linear triangle thermo-mechanical elements.

The length and height of the work piece were 8 mm × 1.61 mm. A horizontal displacement, corresponding to the cutting speed times the time interval of interest, was applied to the nodes at the right side of the tool (the tool motion is from the right to the left). The left side and the bottom side of the work piece was fixed. The particle connectivity using *Delaunay triangulation* was updated every 8th time step increment, equivalent to a tool displacement of 0.0032 mm. The minimum distance between particles used in the numerical simulations was 0.012 mm. More information about meshing parameters is provided in Table 2. The effect of insertion of particles is illustrated in Figure 8.

Material properties for the work piece material are shown in Table 1, Table 3, Table A1 and Table A2. Material properties of the tool are assumed thermo-elastic and are shown in Table 4.

The model for tool–chip interface employed in this study is a regularized Coulomb friction law. A value for the friction coefficient was μ=0.5 (see Rodríguez et al. [28]) and the regularization parameter $V_{rel}$ = 10 mm/s. The regularized Coulomb friction law improves the convergence rate of the NR algorithm, this is one of the reasons to use a regularized Coulomb friction law.

In this study, the fraction of plastic work converted into heat is assumed to be constant and equal to 90%, following the standard assumption of the FEM modeling of metal cutting process [7,31]. The heat generated due to friction is also calculated and applied as a surface heat flux, 50% of the heat generated due to friction goes to the tool and the other 50% goes to the work piece. The contact heat transfer coefficient used is $5\times 10^{6}\frac{W}{m^{2}C}$ according to [31,32].

The predicted cutting forces and chip shapes are strongly influenced by the minimum distance between particles and as well as the maximum time step used in the numerical simulation. For that reason, we select experiment number 4 (see Table 5) with the JC model with materials properties 7 to select the minimum particle size to be used in the primary and secondary shear zone and the maximum tool displacement per time step to be used in the numerical simulations. A tool displacement per time step of 0.0032 mm and a distance between particles of 0.012 mm converges to a value of cutting force 1215 N and a feed force of 720 N. An explanation of the meaning of cutting and feed forces in machining processes is given in [84,85]. The convergence analysis of forces and chip shape was done with a converge tolerance in forces of $1\times 10^{-5}$. The minimum particle distance, the tool displacement and convergence tolerance obtained with experiment number 4 were used in all the numerical simulations used in the present work. The computing time for experiment 4 with JC model with materials properties 7 takes 1 h and 40 min running on a Dell computer with an Intel Xeon(R) CPU E5-1660 v3 @ 3.00GHz x 16 processor and 32 GB of RAM (Located Lulea, Norrbotten. Sweden). The computer time of experiments with feed 0.05 mm and JC model halves the computing time. The previous examples with DD model increase by approximately 1.3 times.

## 6. Examples

Results from PFEM simulations of orthogonal cutting using a dislocation based constitutive model and Johnson-Cook constitutive model were compared with experimental measurements of cutting and feed forces, see Table 5, Table 6 and Table 7.

### 6.1. Forces

Table 5 presents the measured values of cutting and feed forces at different cutting feeds and different velocities. The values reported are the values at steady state. The reported values include the mean/average and the standard deviation for each of the experiments.

The predicted histories of cutting force $F_{c}$ and feed force $F_{f}$ for the experiment numbers 2, 4 and 6 are shown in Figure 9 and Figure 10, where experiment number 2, 4 and 6 where randomly selected for comparison. The results presented in Figure 9 were obtained with the Johnson-Cook constitutive model and the parameters obtained in this work. The results presented in Figure 10 were obtained with the dislocation density constitutive model. The force values presented in Table 6 and Table 7 were evaluated by averaging the force in the steady state region. Average values of the computed forces in the steady state region are compared with the experimental results in Table 6 and Table 7. The error used for the evaluation of the computed results is defined as
(22)$$error\left(\%\right)=\frac{|F_{i,computed}-F_{i,measured}|}{F_{i,measured}}\times 100,$$
the *i* takes value of *c* and *f*, where *c* and *f* make reference to cutting and feed force respectively.

### 6.2. Chip Geometry

The chip formation was simulated as a continuous process without taking into account any damage criteria. The chip geometry was compared with simulated results achieved by the DD model and by the JC model. A detailed comparison with the two constitutive models is presented in Figure 11 for a feed of 0.05 mm and a cutting speed of 60 m/min, and for a feed of 0.15 mm and a cutting speed of 60 m/min in Figure 12.

In Figure 13, a comparison of the predicted chip shapes at different cutting speeds using the JC model and the material properties 7 is shown. In Figure 14, a comparison of predicted chip shapes at different cutting speeds using DD model. Figure 14 only include results obtained for 30 m/min and 60 m/min cutting speed, because with 120 m/min cutting speed the reached temperature in the work piece is greater than the maximum temperature where DD model material properties are available.

In this work, no experimental measurements of chip thickness and contact length were made. Therefore, only a comparison of simulated chip thicknesses and shear plane angle is presented in Table 8 for a feed of 0.15 mm and a cutting speed of 60 m/min.

### 6.3. Material Response

The Figures presented in this section correspond to the steady state conditions. The results shown are for the cutting speed of 60 m/min and feed of 0.15 mm (Experiment 4). Results from the DD and the JC models are presented.

Temperature fields are presented in Figure 15. Maximum temperature takes place in the contact between the chip and rake face of the tool. Figure 15a shows the temperature field generated by the DD model with a maximum value of 1298.1 K, temperature distribution related to the JC model with material properties having a maximum of 1562.3 K is shown in Figure 15b.

Figure 16 illustrates the distribution of effective plastic strain rates in the primary and secondary shear zones. Figure 16a presents a maximum plastic strain rate value of 61,426 s${}^{-1}$ calculated by the DD model. The distribution of plastic strain rates having a maximum value of 37,750 s${}^{-1}$ calculated using the JC and material properties 7 is shown in Figure 16b. It is important to remark that Figure 16a,b present the contour field at the same instant of time, the difference in the chip shape is a consequence of the different models.

The vacancy concentration and the dislocation density are shown in Figure 17. Close to the tool rake face and over the machined surface (see Figure 17) a significant increase of vacancy concentration coupled with the dislocation recovery is present. In the external surface of the formed chip and a short distance below the surface the maximum value of dislocation takes place, the increased dislocation density controls the hardening of the material (see Equations (Equation 4) and (Equation 5)).

## 7. Discussion

Machining is a very important process in the manufacturing industry. The ability to predict process parameters and final properties from the process with numerical modelling is an important challenge. Good numerical tools that virtually can reproduce the machining process will save time and improve final product quality. The goal of the present study was to compare the predictive capability of a DD model with the JC model in the numerical modeling of orthogonal cutting processes. The comparison was made in terms of predicted cutting forces, shear angles and deformed chip thicknesses.

We can state that the PFEM method is a particular class of Lagrangian formulation based on the strengths of the FEM and particle methods. For example, as the PFEM is based on FEM, PFEM converges to the solution by decreasing the distance between particles. Also, PFEM uses ingredients from particle methods and has the ability to predict large deformations, fracture or separation of material from the main domain. The goal is to model discontinuous chip and the PFEM give possibility of model large deformations robustly with high accuracy. In this work we do not consider fracture or separation of material from the main domain. However, these phenomenon can readily be included within the PFEM modelling framework.

One of the main difficulties of modelling metal cutting processes with FEM is the availability of a robust remeshing scheme which includes an accurate transfer of historical variables from mesh to mesh. The PFEM solves the problems of remeshing and diffusion of state variables trough the use of *Delaunay triangulation* to update the particles connectivities in the updated configuration among other ingredients that were introduced in Section 2.

Analysing the results, Figure 9 and Figure 10 show that for a feed of 0.15 mm, the cutting forces decrease when cutting speed is increased, following the same tendency presented by experimental results in Table 5. Furthermore, Table 5 shows that the standard deviation of the measured forces is less than 40 N, representing less than the 7% of the mean forces in all the experiments. Also, Figure 9 and Figure 10 show that with the JC model with material properties number 7, the influence of cutting speed on feed forces cannot be identified. Meanwhile, the DD model predict a decrease in feed forces with the cutting speed.

The JC model with material properties number 3 and 4 better predict the cutting forces with an error of the order of 7%. Meanwhile, the JC model with material properties number 5 gives an error of the order of 80%, being greater for a small feed (see Table 6 and Table 7). The JC model with our parameters predict the cutting force with a error of the order of 40%. The DD model for a cutting speed of 60 m/min and feed of 0.15 predicts the cutting force with an error of 6.3% and the feed force with an error of 0.4%. For a smaller feed, the error in the cutting force and DD model increase up to 27.1%. The JC with material properties number 1 and the DD model better predict the feed forces. The differences between the numerical results and experiments can be related to the fact that the no strain softening or damage was included and that the friction at the tool chip interface was modeled using a regularized Coulomb friction law [86,87,88]. The average error for experiment 3 with the JC model for the cutting force is 37.4% and for the feed force is 16.1%, meanwhile for the same experiment the error in the cutting with the DD model is 27.1% and for the feed force is 2.1%. For experiment 4, the average error in experiment 3 with the JC model in the prediction of the cutting forces is 31.2% and the error in the feed force is 11.7%, meanwhile the error for the DD model are 6.3% and 0.4%, respectively. The above results show that the DD model when compared with the JC model with 7 sets of materials properties is better in the prediction of the cutting and feed forces. Figure 11 and Figure 12 present the predicted chip shapes for the JC model and the DD model for experiments number 2 and 4. In comparison with other numerical simulations, the least similar chip is the one predicted with JC model and properties number 5. The predicted chip with the DD roll faster in comparison with the other predicted chips. One possible explanation of the predicted chip with the DD model is that DD model predict a higher thermal softening of the material in the primary and the secondary shear zone (the strain rates and strain in the primary shear zone are higher than the maximum values with which the DD model was calibrated).

In Table 8, a comparison of the predicted chip thickness and predicted shear angle for the JC with 7 different sets of parameters and the DD model is presented. The predicted results shows that the JC with materials properties number 5 predict the larger chip thickness and the DD predict the smaller value of chip thickness. The results in Table 8 present that the JC with material properties 5 predict the smaller shear angle, the other simulations predict a shear angle close to 28°. The predicted chip thickness with JC and materials properties 5 is out the order in comparison with the predicted values from other simulations. The predicted chip thickness and shear show the same tendency seen previously with the predicted cutting forces and chip shapes.

Figure 13 and Figure 14 show that an increase in the cutting speed results in a chip that roll faster, the same tendency is predicted both by the JC model and the DD model.

The results in Reference [89] show that errors below 10% indicate a good fit between numerical simulations and experiments. In this work, a good precision is obtained and the errors predicted with the DD for Ti6Al4V are similar to the ones presented for AISI 316 L steel using FEM in References [31,32] and using PFEM [90]. Furthermore, in terms of implementation inside a PFEM code, the DD is more difficult to implement because it need to solve an extra set of coupled ordinary differential equations at elementary level that describe the evolution of dislocations density and vacancy concentration. Also, the JC model needs 5 parameters meanwhile the DD model needs all the set of parameters presented in Table A1 and Table A2. At the same time, according to Reference [40] the DD is more difficult to calibrate because it involves a set of coupled differential equations meanwhile the JC is a power law. In the opinion of the authors of this work, the DD model should be used when a detailed and accurate estimation of the process of chip formation is required, while the JC can be used when a gross estimate is required. In addition, one advantage of DD models over the JC models is that if different users calibrate the model with the same set of experimental stress-strain curves, the parameters of the DD model obtained will be the same [40]; this is due to the fact that in the DD model the evolution of the strain-stress curve is described by means of an ordinary differential equation (ODE).

The prediction of dislocation density allows for the study of the material hardening/softening taking place due to the increase/decrease of the long-range component of the flow stress. Having access to the value of the short-range and the drag components of the flow stress allows us to identity which mechanics of hardening/softening are active in the material. In the JC model it is only possible to know if the material is in hardening or softening. Another comparison is possible when the evolution of dislocation density field is measured; it can be contrasted with the prediction of numerical results. This allows the identification of which one of the mechanisms has a major contribution to the increase/decrease of the flow stress.

## 8. Conclusions

Two constitutive models of Ti6Al4V were implemented within a PFEM framework and were compared for the simulation of a metal cutting process. The constitutive models were the DD and the JC with a 7 sets of parameters available in the literature. We believe that the DD model is more physically correct than the JC model, not only because it accurately predicts closely cutting and feed forces, but because the DD model is based on the underlying physics of deformation. Furthermore, the DD model is not a power law that try to fit the complex stress-strain behaviour of Ti6Al4V at high strain, high strain rates and high temperatures. These findings place the PFEM in combination with a DD model as an excellent candidate for modeling machining problems.

The results presented in this work show how the PFEM solves the problems of mesh distortion and diffusion, typical of standard FEM, placing the PFEM in a competitive position with the FEM. Also, the PFEM has not only shown that it solves the numerical problems of the FEM, it has shown that it allows us to obtain precise, accurate and robust results in industrial applications of metal cutting. A literature review allows us to conclude that, among the new numerical methods developed to simulate realistic cutting, the only one that is capable of modeling cutting with dislocation-based models is the PFEM. The other numerical methods need extra development and improvement. A detailed explanation of why the PFEM is the only new method to model realistic cutting processes with a DD model is presented in References [6,7]. Among the main limitations of the new methods is that they usually consider rigid and isothermal tools, and often neglect the friction between the chip and the tool.

## Figures and Tables

**Figure 1 materials-13-01979-f001:**
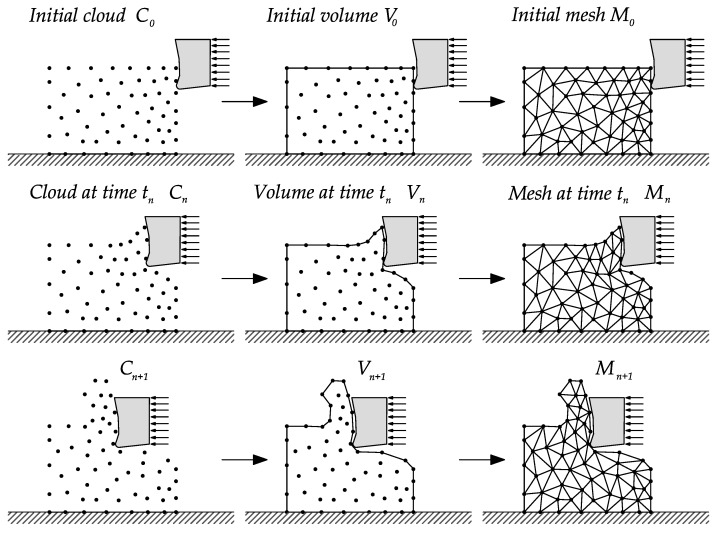
Remeshing steps in a standard Particle Finite Element Method (PFEM) numerical simulation.

**Figure 2 materials-13-01979-f002:**
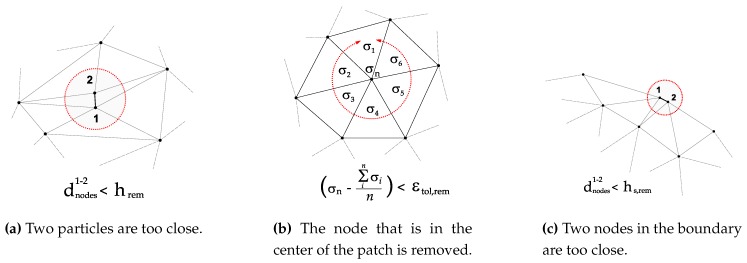
Three main criteria to remove a particle.

**Figure 3 materials-13-01979-f003:**
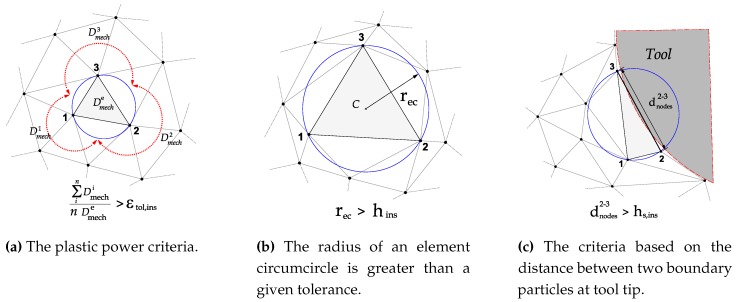
Three main criteria to add a new particle.

**Figure 4 materials-13-01979-f004:**
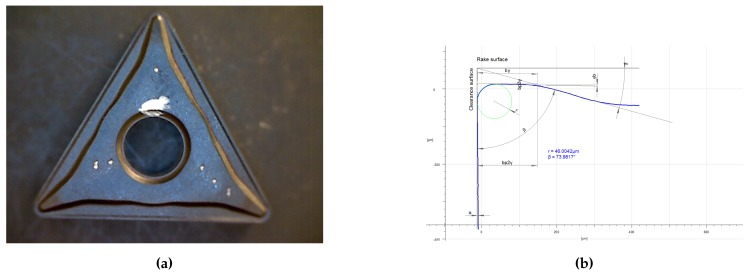
Experimental tool; (**a**) insert, (**b**) cutting edge profile.

**Figure 5 materials-13-01979-f005:**
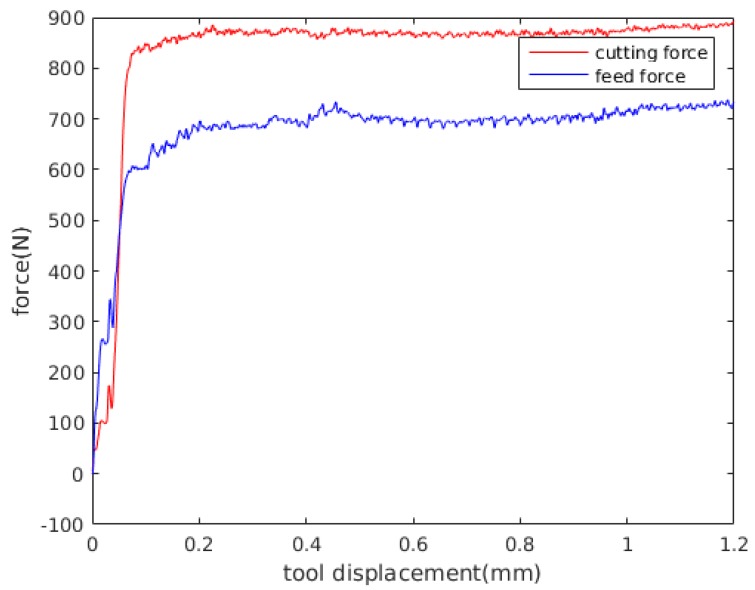
Experimental forces vs. tool displacement for experiment 4 (see Table 5).

**Figure 6 materials-13-01979-f006:**
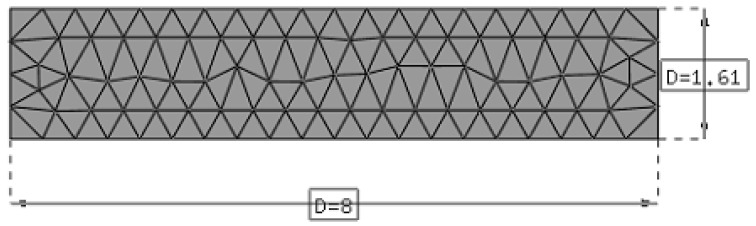
Mesh used to create the particles at the beginning of the simulation. Feed 0.05 mm. Dimensions are in mm.

**Figure 7 materials-13-01979-f007:**
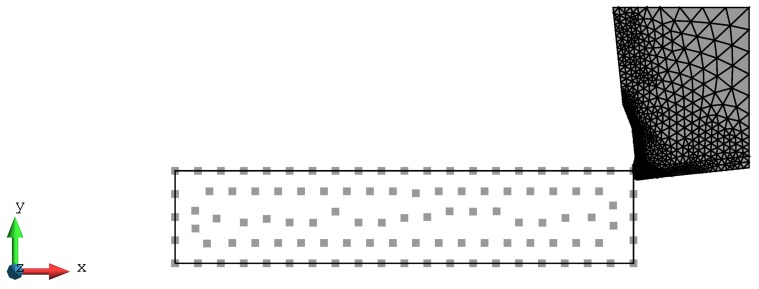
2D plane strain PFEM model of orthogonal cutting: initial set of particles. The picture is for a feed of 0.05 mm.

**Figure 8 materials-13-01979-f008:**
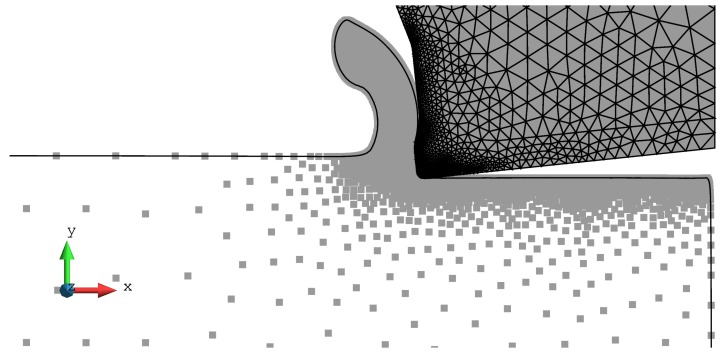
Particles used at the end of the numerical simulation. The Johnson-Cook constitutive model was used with the parameters with parameters 7.

**Figure 9 materials-13-01979-f009:**
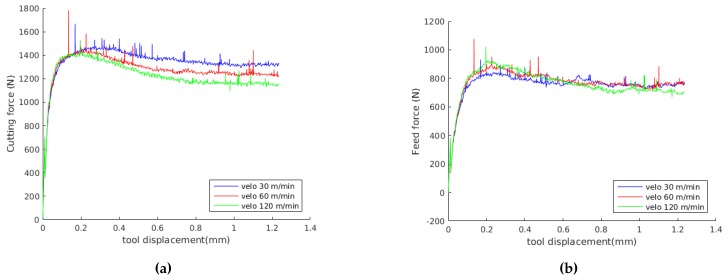
Predicted forces using the Johnson-Cook model with our parameters 7 (see Table 1); (**a**) cutting, (**b**) feed.

**Figure 10 materials-13-01979-f010:**
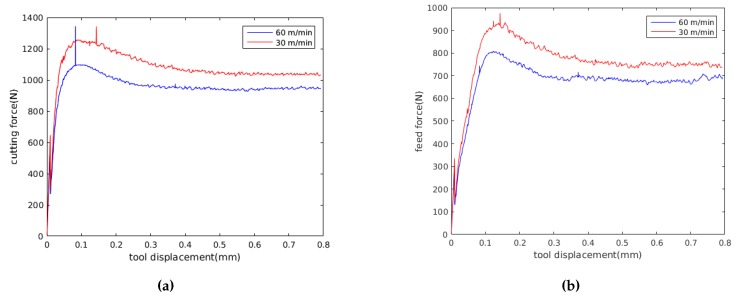
Predicted forces using the dislocation density model; (**a**) cutting, (**b**) feed.

**Figure 11 materials-13-01979-f011:**
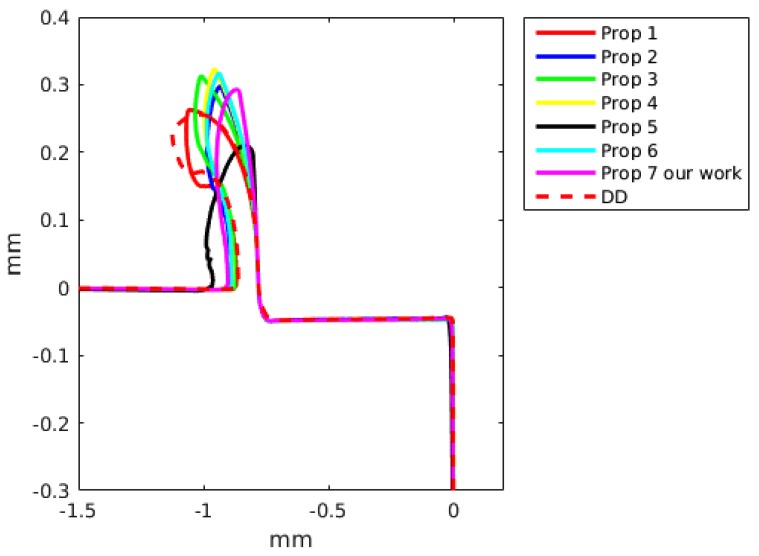
Predicted chip shape using two constitutive models: dislocation density (DD) and Johnson-Cook. Feed of 0.05 mm and a cutting speed of 60 m/min (Experiment 2).

**Figure 12 materials-13-01979-f012:**
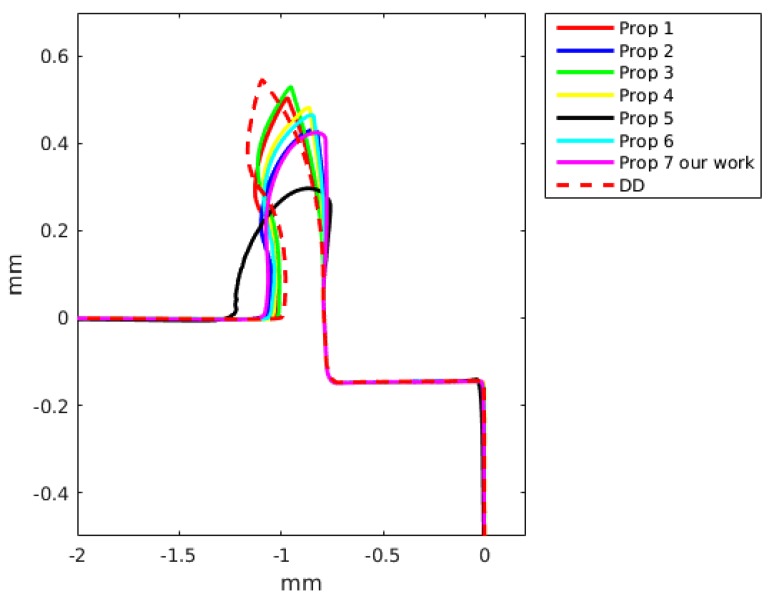
Predicted chip shape using two constitutive models: dislocation density (DD) and Johnson-Cook. Feed of 0.15 mm and a cutting speed of 60 m/min (Experiment 4).

**Figure 13 materials-13-01979-f013:**
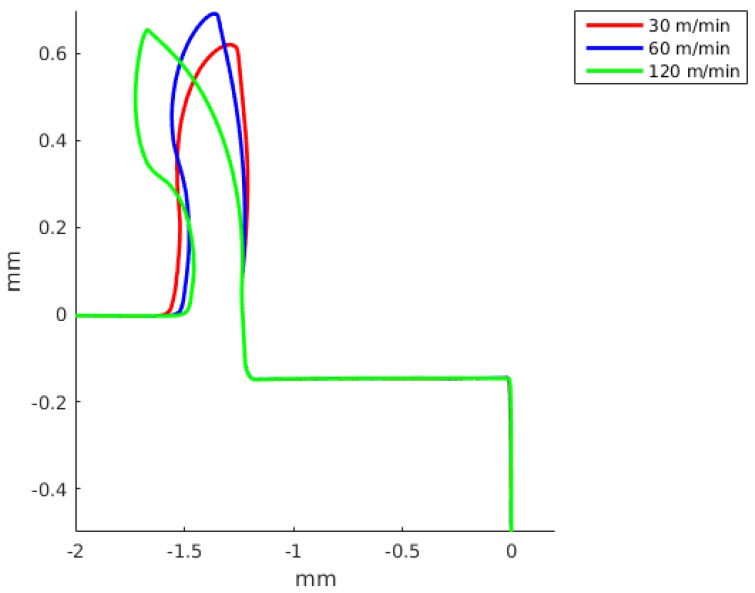
Predicted chip shape at different cutting speeds using Johnson-Cook model and material properties 7 (see Table 1). Feed of 0.15 mm.

**Figure 14 materials-13-01979-f014:**
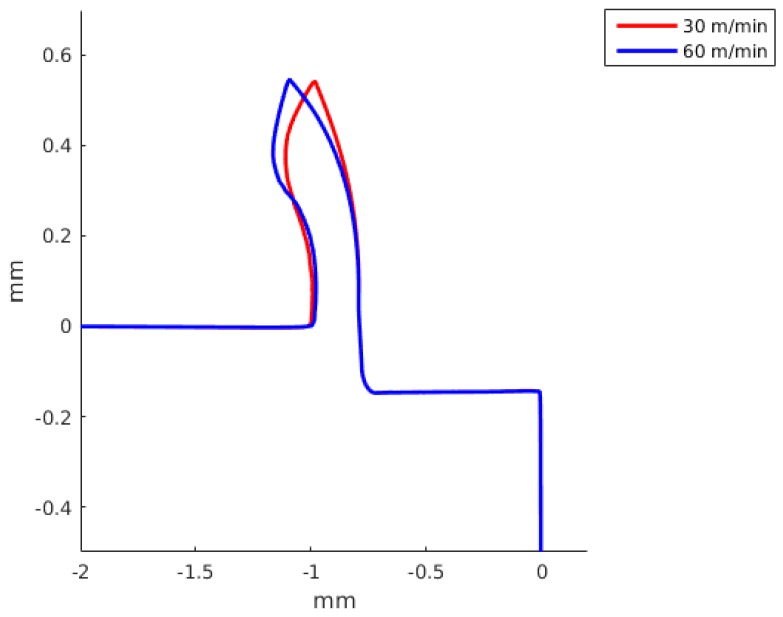
Predicted chip shape at different cutting speeds for the dislocations density model. Feed of 0.15 mm.

**Figure 15 materials-13-01979-f015:**
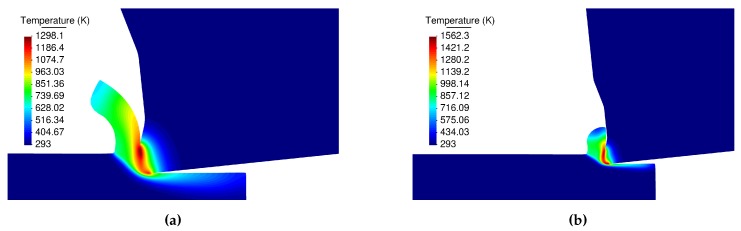
Temperature field; (**a**) the dislocation density, (**b**) the Johnson-Cook model.

**Figure 16 materials-13-01979-f016:**
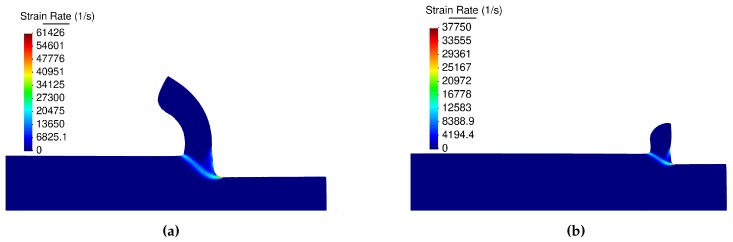
Plastic strain rate; (**a**) the dislocation density, (**b**) the Johnson-Cook model.

**Figure 17 materials-13-01979-f017:**
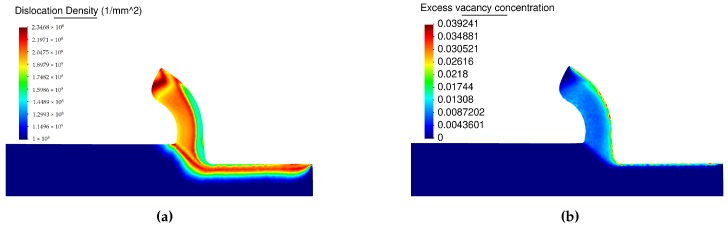
State variables obtained from the dislocation density (DD) model; (**a**) dislocation density, (**b**) vacancy concentration.

**Table 1 materials-13-01979-t001:** The parameters of the Johnson-Cook constitutive equation for Ti-6Al-4V.

Set	*A* (MPa)	*B* (MPa)	*n*	*C*	*m*	${\dot{\varepsilon }}_{0}$ (1/s)	Reference
1	782.7	498.4	0.28	0.028	1	$1\times 10^{-5}$	Lee and Lin [61]
2	724.7	683.1	0.47	0.035	1	$1\times 10^{-5}$	Lee and Lin [62]
3	968	380	0.421	0.0197	0.577	1	Li and He [63]
4	862.5	331.2	0.34	0.012	0.8	1	Meyer and Kleponis [64]
5	1098	1092	0.93	0.014	1.1	1	Chen et al. [65]
6	997.9	653.1	0.45	0.0198	0.7	1	Seo et al. [66]
7	860	612	0.78	0.08	0.66	1	Our work

**Table 2 materials-13-01979-t002:** Mesh parameters used in the numerical simulations.

Parameter	Size
$h_{rem}$	0.012 mm
$h_{ins}$	5 times $h_{rem}$
$h_{s,rem}$	0.012 mm
$h_{s,ins}$	5 times $h_{s,rem}$
$\epsilon_{tol,rem}$	Problem dependent (remove a particle if $r_{ec}$ < 5 × 0.012 mm)
$\epsilon_{tol,ins}$	Problem dependent (insert a particle if $r_{ec}$ > 0.012 in the center of the element. In case of boundary elements insert it in the boundary side)

**Table 3 materials-13-01979-t003:** Thermo physical properties of Ti6Al4V used in the simulations with the Johnson-Cook and dislocation density model (the temperature is given in Kelvins).

Property	Equation	Units	Reference
Young’s modulus *E*	$107\times 10^{3}-200\times T\cdot \exp\left(-\frac{1300}{T}\right)$	MPa	Babu [40]
Thermal conductivity $k_{T}$	0.015(T−273.15)+7.7	W/m K	Karpat [83]
Poisson’s ratio ν	0.33	−	−
Heat capacity $C_{p}$	2.7exp(0.0002(T−273.15))	N/mm${}^{2}$K	Karpat [83]

**Table 4 materials-13-01979-t004:** Thermo physical properties of the tool.

Property	Value	Units
Young’s modulus *E*	$540\times 10^{3}$	MPa
Thermal conductivity $k_{T}$	25	W/m K
Poisson’s ratio ν	0.3	−
Heat capacity $C_{p}$	2.12	N/mm${}^{2}$K

**Table 5 materials-13-01979-t005:** Experimental results. std represents the standard deviation.

Experiment	Cutting	Feed	Cutting	Feed	std	std
Number	Velocity (m/min)	(mm)	Force (N)	Force (N)	Cutting Force (N)	Feed Force (N)
1	30	0.05	420	486	4.35	7.91
2	30	0.15	940	760	13.46	20.50
3	60	0.05	399	478	3.51	9.31
4	60	0.15	888	689	4.12	9.13
5	120	0.05	440	528	17.73	40.03
6	120	0.15	840	760	4.62	11.61

**Table 6 materials-13-01979-t006:** Numerical results at 60 m/min and a cutting depth feed of 0.05 mm (Experiment 3). Error is measured with respect to experimental results presented in Table 5 using Equation (Equation 22). DD is an abbreviation for the dislocation density model.

Material	Cutting	Feed	Error	Error
Properties	Force (N)	Force (N)	Cutting (%)	Feed (%)
1	539	476	35.1	0.4
2	620	500	55.4	4.6
4	373	315	6.5	34.1
5	765	569	91.7	19
6	475	396	19	19.2
7	588	446	47.4	6.7
DD	507	468	27.1	2.1

**Table 7 materials-13-01979-t007:** Numerical results at 60 m/min and a feed of 0.15 mm (Experiment 4). Error is measured with respect to experimental results presented in Table 5 using Equation (Equation 22). DD is an abbreviation for the dislocation density model.

Material	Cutting	Feed	Error	Error
Properties	Force (N)	Force (N)	Cutting (%)	Feed (%)
1	1157	721	30.3	4.6
2	1303	768	46.7	11.5
3	819	546	7.8	20.8
4	821	536	7.5	22.2
5	1570	758	76.8	10.0
6	1003	628	13	8.9
7	1215	720	36.8	4.5
DD	944	692	6.3	0.4

**Table 8 materials-13-01979-t008:** Predicted chip thickness and shear angle at 60 m/min and a feed of 0.15 mm (Experiment 4). DD is an abbreviation for dislocation density.

Material	Chip	Shear
Properties	Thickness (mm)	Angle (deg.)
1	0.207	27
2	0.229	26
3	0.206	28
4	0.224	28
5	0.320	18
6	0.230	27
7	0.253	25
DD	0.194	30

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
