# Peer review of "Dislocation Density Based Flow Stress Model Applied to the PFEM Simulation of Orthogonal Cutting Processes of Ti-6Al-4V"

_materials, 2020, doi:10.3390/ma13081979_

Round 1

Reviewer 1 Report

In this paper “Dislocation density based flow stress model applied to the PFEM simulation of orthogonal cutting processes of Ti-6Al-4V,” written by Juan Manuel Rodríguez, Simon Larsson, Josep Maria Carbonell and Pär Jonsén, the Particle Finite Element Method together with a dislocation density constitutive model are introduced to simulate the machining of Ti-6Al-4V.

The reviewer cannot agree with a publication of this article in MDPI Materials with the current form. I have several concerns which should be addressed before the paper is acceptable. I note them below:

  1. Reviewer believes that Section 2. The Particle Finite Element Method was already given many times through the authors’ previous works. For example, Figures 1-3 and Box 1 were presented in Comput Mech (2018) 61:639–655. This section can be largely reduced.
  2. Literature review. As the authors mentioned in the abstract, metal cutting involves high strain rates and high temperatures, which may cause the dynamic strain aging phenomenon. Recently, Voyiadjis and his co-workers developed the physically based dislocation density constitutive model considering the dynamic strain aging. Check out the following works:
    Voyiadjis, G. Z., Song, Y., & Rusinek, A. (2019). Constitutive model for metals with dynamic strain aging. Mechanics of Materials, 129, 352-360.
    Voyiadjis, G. Z., & Song, Y. (2020). A physically based constitutive model for dynamic strain aging in Inconel 718 alloy at a wide range of temperatures and strain rates. Acta Mechanica, 231(1), 19-34.
  3. In line 277, epsilon dot should be epsilon dot^p. Please also check if ln (epsilondot^ref/epsilon dot^p) in Eq. 8 is correct. Reviewer thinks it is upside down, i.e. it should be ln (epsilondot^p/epsilon dot^ref).
  4. Typo in line 289. Acoording.
  5. Numerical results from PFEM using a dd constitutive model and JC model were validated by comparing with the experiments. Please provide the information about the experiments.
  6. Errors indicated in Tables 5 and 6 are huge. Why?
  7. What does x-axis in Figure 6 stand for? Units are also missing.
  8. What is DBM in Figure 8? What do x-axis and y-axis in Figures 8-11 stand for? Units are missing.
  9. As revealed in Figure 13, high strain rate is resulted in shear bands. Which condition did the authors assume between isothermal condition and adiabatic condition in the model and FEM?
  10. Different feed may affect the numerical results, for example width of shear bands or force-displacement curves. Could the authors check this out?

Reviewer 2 Report

English check
88 comparison. The the adaptation of the use of a DD...
270 where, σG is the athermal stress contribution...
272/273 is the inmobile dislocation density...

Further clarification
1
314 An orthogonal cutting operation was employed to mimic 2D plane strain conditions. The depth of the cut, used for all test cases is equal to 3 mm.
vs
323 The dimension of the work piece in the FEM model was 8 × 1.61 mm.

2
Further details should be provided for clear definition of Feeding Force and Cutting Force 

3
Even if a very detailed presentation of the PFEM is given, no reference to the experimental work is made. The paper would benefit a brief description of the experimental setup, measurement details and confidence, etc.

Reviewer 3 Report

Abstract

The authors mentioned that DD model is based on physics and ‘Improving the results obtained with phenomenology based models like the Johnson Cook model.’ I have some doubt on that. It can be seen that DD model solves the stress by three parts based on physics, but a lot of calibration parameters are included to calculate each component. This calibration process is in nature as same as phenomenology based models. Please discuss about that.

Section 1

1.Page 2, line 60, ‘With the advantage that the computational cost of the PFEM is similar to that of specialized FEM programs [62].’ It is hard to believe that PFEM could remain the computational efficiency with all these additional steps involved in remeshing. There is no computation time reported throughout the paper. It is critical to include this information for all simulations.

2.Page 2, line 66,’ This numerical method is the the better candidate for the purposes of this work.’ There is a typo with additional ‘the’.

3.Page 3, line 77, ‘one of the weakness of these empirical relationships is that they have not predictive capabilities outside the experimental calibration range.’ However, on page 17, section 5.2, ‘Figure 11 only include results obtained for 30 m/min and 60m/min cutting speed, because with 120 m/min cutting speed the reached temperature in the work piece is greater than the maximum temperature were DD model material properties are available.’ It seems that DD model has more limitation on the prediction range than J-C model, please comment on that. Also, there is a typo, it should be ‘where DD model material properties are available’

Section 2

  1. Page 5, line 176, 177, and Figure 2(c) Figure 3 (c), it seems that particle is removed when distance smaller than hs, and added when larger than hs. Does it mean that there is no equilibrium state throughout the simulation unless the distance between nodes is exactly hs?
  2. What is the exact value of εtol and hs in simulation? Above figure 5, it mentioned a minimum distance between particles of 0.012 mm, does it relate to any removal or addition criterion?

Section 3

  1. Section 3.2 is difficult to track; a flowchart is recommended to clearly show the relationship between equations.
  2. For section 3.2.1, is it an iterative process to calculate ρi in eq. (6)? ρi is calculated from eq. (14), (16), (17), (18), where ρi itself is used in these calculations. If it is iterative, what are the initial value and stopping criteria?
  3. For eq. (18), it says that the reduction of dislocations density due to globularization is zero if ρi < ρicrit. However, eq. (18) is zero when ρi = ρeq. Does it mean that ρicrit= ρeq ? If not, does it mean that there will be a jump of reduction rate, when ρi becomes larger than ρicrit?
  4. For section 3.2.2, what is the purpose of eq. (22)? Is it used to calculate Cv in eq. (17)? Again, I understand that some calculations are explained in other references. But whatever presented in the paper, a flowchart or more clarifications are needed to provide a better readability.

Section 4

  1. How is the position of initial particles determined in Figure 4? It seems that particles are randomly distributed inside the work piece.
  2. Page 14, line 347, the authors mentioned that ‘a value of cutting force 558 N and a feed force of 468 N’ based on ‘experiment number 4 (see Table 4) with the JC model with materials properties 7’. However, in table 6, the cutting force is 1215, while feed force is 720. Please clarify.

Section 5

  1. Section 5.1, the authors mentioned that ‘The values reported are the values at steady state’. How is the steady state determined? Could the authors demonstrate that in Figure 6 and 7?
  2. Deviation of measurements and predictions needs to be added in table 4-6.
  3. The experimental process needs to be described in more detail. What is the experimental setup (Add a figure is preferred)? What are the parameters of the tool? And how is the force measured?
  4. It would be better to provide a measured force signal over time besides figure 6 and 7.
  5. The horizontal axis title is missing in Figure 6.
  6. Section 5.2 and 5.3 have no experimental data, which makes the comparison between DD and JC model meaningless. In the abstract, the authors claim that ‘The dislocation density model, although it needs more thorough calibration, shows an excellent match with the results.’ This statement is invalid without more experiments. If additional measurements are impossible, the authors may consider doing comparison with literature.
  7. Page 23, line 413, ‘In this work we do not consider fracture or separation of material from the main domain.’ Is this supported by experiment? Is the chip continuous and not separated from work piece during experiment?

Round 2

Reviewer 1 Report

The authors have put a lot of effort to modify the manuscript. This work can be accepted with the current form.

Author Response

Thank You very much for your feedback, comments and the time you invested reviewing our paper.

Reviewer 3 Report

Section 1

Old comment 'Page 3, line 77, ‘one of the weakness of these empirical relationships is that they have not
predictive capabilities outside the experimental calibration range.’ However, on page 17, section
5.2, ‘Figure 11 only include results obtained for 30 m/min and 60m/min cutting speed, because with
120 m/min cutting speed the reached temperature in the work piece is greater than the maximum
temperature were DD model material properties are available.’ It seems that DD model has more
limitation on the prediction range than J-C model, please comment on that. Also, there is a typo, it
should be ‘where DD model material properties are available’
The typo was fixed.
Our comment is based on the predicted stress at high strain and high strain rates, because the
predicted stress with DD is better than the best prediction with the different sets of Johnson Cook
parameters. The better behavior of the DD model comes from considering the underlying physics of
the materials at the conditions of interest. For comparison, we include the average errors in the
predictions of the cutting and feed forces for the JC model, this average shows clearly that the DD
model is better in the prediction of the cutting and feed forces.
'
New comment: The authors' reply didn't answer the question. This isn't about the accuracy of the model, but the prediction range. If JC model can predict 30 to 120 cutting speed, but DD model is only valid between 30 and 60. Then it is not right to say that ' the weakness of JC model is lack of predictive capabilities outside the range'

Section 3

Old comment:' For eq. (18), it says that the reduction of dislocations density due to globularization is zero
if
ρi< ρicrit. However, eq. (18) is zero when ρi = ρeq. Does it mean that ρicrit= ρeq
? If not, does it
mean that there will be a jump of reduction rate, when
ρi becomes larger than ρicrit
?
There was a typo in the equation and a misunderstanding in the paragraph next to it. That
term is active when
ρi>=ρicrit. Until ρ< ρeq. '
New comment: The explanation of ρicrit  needs to be kept in the manuscript.

Section 5

Old comment:'2. Deviation of measurements and predictions needs to be added in table 4-6.
In the caption of Table 5 and 6 we clarify that the error was measured against experiments
in the lab.
'

New comment: The authors' reply didn't answer the question. This comment is about deviation instead of error. Deviation means that the measured or predicted forces signals are not constants, even at steady state, and this deflection of force signal needs to be added besides the average.
